# The Confidence of and Concern about Using Mobile Banking among Generation Z: A Case of the Post COVID-19 Situation in Thailand

Wischaya Silanoi [1], Phaninee Naruetharadhol [2] and Khwanjira Ponsree [2,*]

[1] Office of Learning Promotion and Provision of Academic Services, Valaya Alongkorn Rajabhat University under the Royal Patronage, Pathum Thani 13180, Thailand
[2] International College, Khon Kaen University, 123, Moo 16, Mittraphap Rd., Nai-Muang, Muang District, Khon Kaen 4002, Thailand
* Correspondence: khwanpo@kku.ac.th

**Abstract:** This research paper focuses on mobile banking acceptance among Generation Z users by combining the technology acceptance model (TAM) and the unified theory of acceptance and use of technology (UTAUT) theory, as well as extending the theories with perceived trust and risk. During the COVID-19 pandemic, the online questionnaire survey was distributed using Google Forms, and the sample group was Thai Generation Z who were aged between 18 and 25 years old. The research objectives aim to (i) investigate the crucial effects that potentially affect user intention and actual usage of mobile banking, (ii) identify the most influential factor impacting users' intention and behavior, (iii) further study the trust and risk perception of Generation Z users on mobile-banking intention and actual usage, (iv) discuss the findings with the antecedent studies, and (v) contribute the research findings both theoretically and practically. The proposed constructs include perceived usefulness, perceived ease of use, social influence, facilitating conditions, perceived trust, perceived risk, behavioral intention, and actual usage. There are fourteen proposed hypotheses to be tested. Based on the outcomes and the standardized coefficient beta, perceived usefulness ($\beta = 0.518$) was the strongest factor determining Generation Z's behavioral intention, while perceived ease of use ($\beta = 0.809$) impacting perceived usefulness demonstrated the strongest relationship among all of the hypotheses.

**Keywords:** mobile banking; Generation Z; TAM; UTAUT

## 1. Introduction

Since the COVID-19 pandemic spread worldwide in late 2019, the world faced an economic recession during 2020—the global economy is projected to shrink by over four percent, instead of raising four percent as was previously forecasted (United Nations Conference on Trade and Development 2021a). In contrast, COVID-19 is the key driver and accelerator of digital transformation (Asian Development Bank 2021; World Bank 2020b; World Trade Organization 2020), especially in Thailand (Srisathan and Naruetharadhol 2022), leading to the trend of cashier-less payment in online shopping (Smith and Sammer 2021). The United Nations Conference on Trade and Development (United Nations Conference on Trade and Development 2021b) reported that the pandemic has stimulated the e-commerce market to reach USD 26.7 trillion—raised by four percent from the year 2018 to 2019. Another report from (ADB) Asian Development Bank (2018) demonstrated that, in the recent decade, e-commerce in the Asian business-to-consumer market had the fastest growth in the global e-commerce marketplace, reaching about 4.5 percent of growth relative to the gross domestic product (GDP) in Asia and the Pacific by the end of 2015. Small and medium enterprises are gaining various advantages from e-commerce by entering global markets, e.g., competing on an international scale and expanding their product or

service internationally (Asian Development Bank 2018). Moreover, Asian Development Bank (2021) emphasized that global output, trade, and commerce were directly accelerated by the pandemic. For instance, by 2025, there will be a huge increase in the size of the digital sector, by around 20 percent, over the baseline of the year 2020. The aforesaid growth will impact the global output by USD 21.4 trillion, accumulated over 5 years, or USD 4.3 trillion a year on average over 2021 to 2025, which accounts for 5.4 percent over the 2020 baseline. On the other hand, Asia would gain an economic dividend of around 6.1 percent of the 2020 regional GDB baseline, more than USD 1.7 trillion a year or greater than USD 8.6 trillion accumulated over the 5 years to 2025. The trade volume in Asia is projected to increase about 6.8 percent from the 2020 regional baseline trade, more than USD 1.0 trillion or an accumulated USD 5 trillion in 5 years. The aforementioned reasons for all those growing trends would increase the potential for digital connectivity, productivity, and economic gains, and stronger digital trade services in Asia and the Pacific (Asian Development Bank 2021).

In Thailand, COVID-19 accelerated digital transformation the same as it did globally (Deloitte Thailand 2020; Srisathan and Naruetharadhol 2022). The Thai national strategy, Thailand 4.0, aims to exit the middle-income trap, inequality trap, and imbalance trap. Economic wealth is one of the four dimensions that the strategy focuses on achieving. This dimension focuses on innovation and technology to drive Thai economies and to reduce the dependence on foreign technology (The Secretariat of the Prime Minister 2017). One of the most seen pieces of evidence was the encouragement of digital payment technology and infrastructure by the Bank of Thailand (BOT). The BOT has constructed a standardized digital payment infrastructure and system for the commercial banks in Thailand. This action facilitated Thai commercial banks to further adapt and develop their digital banking, including internet banking and m-banking, resulting in a 74.7 percent (5,609.7 million transactions) rise in volume and a 21.0 percent (THB 54.3 trillion) increase in value from 2018 to 2019 (Bank of Thailand 2019). Therefore, the digital transformation of Thailand has been complementarily accelerated by both the COVID-19 epidemic and Thailand 4.0 strategy.

Regarding the online environment, confidence and concern have continuously risen in the examination for decades (Makki et al. 2016; Okazaki et al. 2012; Liao et al. 2011; McCole et al. 2010; Hanif and Lallie 2021; Kim et al. 2008; Ozturk et al. 2017; Victor et al. 2018; Jahangir and Begum 2008; Migliore et al. 2022; Holland et al. 2003). Recently, Hanif and Lallie (2021) investigated the role of perceived overall cyber security, perceived cyber security risk, perceived cyber security trust, and perceived cyber security privacy on the intention to use mobile banking applications among elderly people aged over 55 years old. However, their outcomes only unveiled a significant effect from perceived cyber security risk on the intention to use mobile banking (Hanif and Lallie 2021). Migliore et al. (2022) conducted cross-national research and concluded that risk barrier has an insignificant effect on behavioral intention to use mobile payment in China and Italy. Victor et al. (2018) addressed privacy concerns as the determinant that potentially shapes consumer behavior in an online dynamic pricing environment. For the present study, confidence and concern regarding mobile banking were represented as perceived trust and perceived risk, respectively.

Generation Z, or shortly called "Gen Z", characterized as digital natives, was born between the mid-1990s and 2010, and has a high digital competence (Oxford Economics 2021). This generational cohort tends to be more independent than their elder cohorts, and is rather interested in startups and early stage companies for their career (Gomez et al. 2018). Moreover, this generation is open to diversities regardless of nationalities, languages, skin colors, genders, and sexual orientations (Gomez et al. 2018). Turner (2015) and Priporas et al. (2017) supported that Generation Z is a heavy user of technology and has a high tendency to accept digital technologies. Furthermore, there is much empirical evidence that focused on Generation Z. For example, there has been a study of Generation Z in Thai culture (Farrell and Phungsoonthorn 2020), a behavioral study of Generation Z

during the COVID-19 situation (Chayomchai 2020; Liu et al. 2021), a study in the context of tourism (Dimitriou and AbouElgheit 2019; Nguyen et al. 2021; Vieira et al. 2020), and a study on mobile payment adoption among Generation Z (Wei et al. 2021). Nevertheless, there was inadequate study of Generation Z focusing on the intention and use of mobile banking post COVID-19 considering trust and risk perception, especially in Thailand. Thus, researchers are willing to fill the aforesaid research gaps. For the research theory, the technology acceptance model (TAM) (Davis 1989) and the unified theory of acceptance and use of technology (UTAUT) (Venkatesh et al. 2003) have been integrated and extended with perceived trust and perceived risk, and some additional path relationships to further explain and discuss the research objectives. Additionally, there are five main research objectives to achieve including: (i) to examine the crucial effects that potentially affect the behavioral intention and actual usage among Generation Z's mobile banking users; (ii) to address the most influential factor that impacts user intention and behavior; (iii) to further investigate the trust and risk perception of Generation Z users on mobile banking intention and actual usage; (iv) to discuss the research findings with the previous studies; and (v) to theoretically and practically contribute based on the recent research outcomes.

To achieve the research objectives and outcomes, the structural equation modeling (SEM) approach was applied to the study (Ullman and Bentler 2003). The SEM approach helps to examine the multiple hypotheses at a single solution, improve the generalizability and reality of the findings, avoid type I error or control the inflation of the experiment, and compute the multiple constructs with numerous measurement items at a time (Buhi et al. 2007). As confirmatory factor analysis (CFA) is a fundamental step of SEM (Anderson and Gerbing 1988), the present study has also assessed the validity of the constructs and measurement items in various aspects. Then, this research paper is organized as follows: Section 2 is the literature review, where the background information on mobile banking and hypothesis development is located. Additionally, methodology is placed in Section 3, where the scope of the study, sample size determination, sampling procedure, data collection process, questionnaire design, and the instrument are demonstrated. The next is Section 4, where research findings and discussion, the demonstration of statistical data, CFA results, SEM findings, and discussion are placed. After that, the research implications are in Section 5. Finally, the conclusion is summarized in Section 6.

## 2. Literature Review

### 2.1. Background of Mobile Banking

Back at the end of the 1990s, mobile banking (m-banking) was begun by the collaboration of German company Paybox with Deutsche Bank launching the first service, firstly developed and tested by European countries such as Austria, Germany, Spain, Sweden, and the United Kingdom, and further expanded and adopted worldwide (Shaikh and Karjaluoto 2015). Mobile personal devices have been utilized for online banking, transactions, or any online payment activities because of their convenience and security. Through an electronic authorization mechanism, the authentication of electronic payment activities, such as online banking, ATM (automated teller machine) transactions, and card-not-present credit/debit card transactions, is secured (Hezerberg 2003). The earliest form of m-banking was an SMS message, which received information on the customer's account balance and was sent by a particular commercial bank (Mallat et al. 2004). Moreover, Mallat et al. (2004) illustrated that the emergence of Java- and WAP-enabled mobile phones (Wireless Application Protocol) utilizing the GPRS (General Packet Radio Services) enhanced the variation of m-banking services. For example, mobile banking has been used for stock trading, fund transfer between bank accounts, and direct payment confirmation process via mobile phone (Mallat et al. 2004). Shaikh and Karjaluoto (2015) reviewed that the expansion of the smartphone user base has had a positive effect on the demand for m-baking services, software houses, and microfinance institutions. A boom in banking services and the number of digital payment service providers delivering innovative services linked with new mobile application designs and new sets of products and services offered allow

their clients to be reached, both banked and unbanked. Commercial banks can gain more market share, enhance operational efficiency, improve customer retention, and provide new employment opportunities. After that, a substantial number of studies, notably on m-banking adoption, have been undertaken under various contextual investigations.

*2.2. Integration of TAM and UTAUT Theories and Hypothesis Development*

There are various theories to examine technology adoption, such as the TAM theory (Davis 1989), the theory of reasoned action (TRA) (Fishbein and Ajzen 1975), the theory of planned behavior (TPB) (Ajzen 1991), innovation diffusion theory (IDT) (Mahajan 2010), and the UTAUT theory (Venkatesh et al. 2003). However, the constructs and the measurement items of the aforesaid theories are similar. For instance, perceived usefulness and perceived ease of use in the TAM theory are similar to performance expectancy and effort expectancy in the UTAUT theory, respectively (Venkatesh et al. 2003). The present study has integrated the TAM theory with the two constructs from the UTAUT theory, namely, social influence and facilitating conditions, and further extended the theories with another two constructs, namely, perceived trust and perceived risk. With the extension, there are eight constructs to measure in the CFA and SEM models, including perceived usefulness (PU), perceived ease of use (PEU), social influence (SI), facilitating conditions (FC), perceived trust (PT), perceived risk (PR), behavioral intention (BI), and actual usage (AU). Therefore, the hypotheses have been developed as the following:

Perceived usefulness is defined as the degree of individual perception towards the usage of technology that would increase a person's job performance (Davis 1989). Perceived usefulness in the TAM theory (Davis 1989) is one of the root constructs of performance expectancy in the UTAUT theory (Venkatesh et al. 2003). In the m-banking acceptance study, numerous scholars supported a significant effect of perceived usefulness on behavioral intention to use m-banking among millennials in Malaysia (Karim et al. 2020) and among bank consumers in Vietnam (Van et al. 2021). Similarly, the effect of performance expectancy in the UTAUT-based theory has been proven by various studies. For example, the moderating effect of experience (Liébana-Cabanillas et al. 2014a) and the moderating effect of age (Liébana-Cabanillas et al. 2014b) on the mobile payment system acceptance among Facebook users have been studied. Their results found that the intention to use was only influenced by the usefulness among experts and did not play a role among inexperienced individuals. On the other hand, the intention to use was positively influenced by usefulness for both people who are aged younger than 35 and older or equal to 35 years old. Furthermore, Liu and Ji (2018) examined the perceived usefulness of online group buying, particularly in the Chinese context. Another study confirmed the positive relationship between performance expectancy towards the behavioral intention among m-banking users in Portugal (Oliveira et al. 2014) and towards the intention to use a 4.5G mobile service in Malaysia (Daniali et al. 2022). Hence, the hypothesis below is proposed.

**Hypothesis 1.** *There is a significant impact of perceived usefulness on Generation Z customers' behavioral intention to use m-banking.*

Perceived ease of use is defined as the perception of an individual towards the easiness associated with using technology (Davis 1989). In the TAM-based theory, there were many studies that revealed the influence of perceived ease of use on behavioral intention, such as customers' decisions towards m-banking usage in Vietnam (Le et al. 2020; Van et al. 2021) and in Malaysia (Karim et al. 2020). Based on the UTAUT-based theory, the antecedent studies also supported the significant relationship between effort expectancy on behavioral intention, such as the study of m-banking adoption among users in Portugal (Oliveira et al. 2014). Furthermore, Liébana-Cabanillas et al. (2014b) and (2014a) further investigated the hidden effect of ease of use on the usefulness. Their findings illustrated a strong effect among this pair of constructs on both age gaps (age <35 and ≥35) and experience groups (expert and inexperienced). Furthermore, many researchers claimed

that perceived ease of use has a significant influence on the perceived usefulness of m-banking in different contextual investigations, e.g., in Malaysia (Karim et al. 2020) and Brazil (Malaquias and Silva 2020). In a similar vein, Saprikis et al. (2022) reported that the effect of effort expectancy on performance expectancy was positively significant for both users and nonusers of m-banking in Greece. Thus, the hypotheses are the following:

**Hypothesis 2a.** *There is a positive significant impact of perceived ease of use on the perceived usefulness of m-banking.*

**Hypothesis 2b.** *There is a positive significant impact of perceived ease of use on Generation Z customers' behavioral intention to use m-banking.*

Social influence is an external factor. Important people (friends, family, and relatives) to individuals can influence their perception of the use of technology (Venkatesh et al. 2003). This construct further developed from the subjective norm (Ajzen 1991, 1989; Davis 1989; Fishbein and Ajzen 1975; Taylor and Todd 1995). The recent researches in various contextual studies have examined the crucial effect of social influence on m-payment adoption behavior among young generations in Taiwan (Wei et al. 2021), on purchase intention of bottled water in the United States (Lin and Xu 2021), and on mobile 4.5G service intention in Malaysia (Daniali et al. 2022). Additionally, the subjective norm from the TAM-based theory also impacted the behavioral intention of Generation Z consumers in the subject of online fashion rental in Vietnam (Pham et al. 2021). Particularly in the m-banking environment, a plethora of studies found that social influence created a repetitive effect on the behavioral intention to use m-banking in Pakistan (Farah et al. 2018), Vietnam (Le et al. 2020), Greece (Saprikis et al. 2022), Maldives (Parayil Iqbal et al. 2022), and Sri Lanka (Samsudeen et al. 2022). Because social influence is an external normative structure that influences an individual's intention and is dependent on friends, family, relatives, and other important people, this might also represent their trust in technology usage (Venkatesh et al. 2003). Thus, the hypotheses were formulated as the following:

**Hypothesis 3a.** *There is a positive significant impact of social influence on the perceived trust of m-banking.*

**Hypothesis 3b.** *There is a positive significant impact of social influence on Generation Z customers' behavioral intention to use m-banking.*

Facilitating conditions are the perception and belief of an individual that there would be the existing resources necessary for them to learn how to use a particular technology and can remove the barrier to use such technology (Venkatesh et al. 2003). Perceived behavioral control (Ajzen 1991; Taylor and Todd 1995) is one of the root constructs of FC and directly affects use behavior (Venkatesh et al. 2003). Antecedent research in different contexts has supported that effect (Davis 1989; Mathieson 1991). When resources are available, individuals can better understand and perceive that such technology is simple to use, thereby developing a favorable attitude and fostering trust among potential users (Gu et al. 2009; Venkatesh et al. 2003). According to Oliveira et al. (2016), if the operational infrastructure exists and it makes mobile payment technology easy to use, the behavioral intention to use such payment technology will increase. In addition to examining the direct impact of facilitating conditions on use behavior, a plethora group of researchers investigated its direct effect on behavioral intention to use m-banking (Farah et al. 2018; Merhi et al. 2021; Owusu Kwateng et al. 2019; Parayil Iqbal et al. 2022; Samsudeen et al. 2022; Saprikis et al. 2022; Thusi and Maduku 2020). Therefore, the hypotheses are proposed as follows:

**Hypothesis 4a.** *There is a positive significant impact of facilitating conditions on the perceived ease of using m-banking.*

**Hypothesis 4b.** *There is a positive significant impact of facilitating conditions on the perceived trust of m-banking.*

**Hypothesis 4c.** *There is a positive significant impact of facilitating conditions on Generation Z customers' behavioral intention to use m-banking.*

**Hypothesis 4d.** *There is a positive significant impact of facilitating conditions on Generation Z customers' actual usage.*

Perceived trust is defined as the degree to which an individual expresses and feels secure or comfortable towards using products or services without concern for any risks or issues (Gefen et al. 2003). Customers' increased trust is essential for digital banking service providers to motivate them to use the service (Page and Luding 2003). A higher level of trust would eliminate the consumer's perceived risk (Fortes and Rita 2016; Jarvenpaa et al. 2000; Oliveira et al. 2014). In this study, perceived trust is the determinant of confidence in mobile banking usage. Many studies have determined the effect of trust on behavioral intention, such as in m-banking adoption among customers (Gu et al. 2009; Oliveira et al. 2014) and a mobile payment system study among Facebook users (Liébana-Cabanillas et al. 2014a, 2014b; Liébana-Cabanillas et al. 2018). Moreover, another study further investigated the relationship of trust on perceived risk (Jarvenpaa et al. 2000; Fortes and Rita 2016). The findings demonstrated a negative effect of trust to risk in an online store study among Australian university students (Jarvenpaa et al. 2000) and in an online purchasing behavior study in Portugal (Fortes and Rita 2016). The relationship between perceived trust and use behavior has also been recently addressed by previous studies, especially in m-banking adoption (Malaquias and Silva 2020; Owusu Kwateng et al. 2019). Thus, the hypotheses of the perceived trust factor were formulated as below.

**Hypothesis 5a.** *There is a negative significant impact of perceived trust on perceived risk.*

**Hypothesis 5b.** *There is a positive significant impact of perceived trust on Generation Z customers' behavioral intention to use m-banking.*

**Hypothesis 5c.** *There is a positive significant impact of perceived trust on Generation Z customers' actual usage.*

Perceived risk is defined as the perception of an individual, whether concern or fear of losing their personal information, confidential information, or transaction information taken by an unauthorized person, that has a negative effect on their perception of electronic services and activities (Fortes and Rita 2016; Glover and Benbasat 2010). Perceived risk had a negative effect on customer intention to use mobile payment in any age range (Liébana-Cabanillas et al. 2014b) and regardless of experience (Liébana-Cabanillas et al. 2014a). Moreover, perceived risk has been demonstrated to have a detrimental impact on customer acceptability of m-banking (Thusi and Maduku 2020). According to the review, we propose the hypotheses of perceived risks as the following:

**Hypothesis 6a.** *There is a negative significant impact of perceived risk on Generation Z customers' behavioral intention to use m-banking.*

**Hypothesis 6b.** *There is a negative significant impact of perceived risk on Generation Z customers' actual usage.*

The antecedent researches of information technology acceptance found that the technology usage was directly influenced by the behavioral intention of an individual in the TAM (Davis 1989), the TPB (Ajzen 1991), and the UTAUT theories (Venkatesh et al. 2003). The evidence of behavioral intention impacting use behavior has been repeatedly proven by many studies, e.g., utilizing UTAUT theory as the base model (Oliveira et al. 2014; Zhou et al. 2010). Farah et al. (2018), Owusu Kwateng et al. (2019), Le et al. (2020), Merhi et al. (2021), and Parayil Iqbal et al. (2022) confirmed a robust impact of behavioral intention on the use behavior and actual usage of m-banking. Hence, the hypothesis was formulated as below:

**Hypothesis 7.** *There is a positive significant impact of behavioral intention on Generation Z customers' actual usage.*

### 3. Research Methodology

*3.1. Scope of the Study and Sampling Procedures*

Generation Z who are aged between 18 and 25 years old in Thailand, considering they are using m-banking services, are the sample group of the present study. This study focuses on this generation because they have the highest digital competence, which is appropriate for an m-banking acceptance study (Turner 2015; Priporas et al. 2017). Furthermore, this similar consumer segment (between 15 and 24 years of age) also has a high literacy rate, about 91.73 percent in 2019 (World Bank 2020a). Especially in Thailand, in 2018 this consumer segment had a literacy rate of 98 percent, which is higher than the global average (World Bank 2020a). Therefore, the researchers perceived this information as an opportunity to explore their behavior regarding m-banking usage. Since the COVID-19 pandemic was exploding at the time of the study, an online questionnaire survey using Google Forms was an appropriate tool to collect the data. The size of the Thai Generation Z population is unknown (National Statistical Office n.d.). Therefore, the sample size calculation by Cochran (1977) was used, with the proportion of the randomized population being approximately 50%, or 0.5, and a reliability of 95%; the recommended sample size was 384.

The data were collected for one and a half months from December 2022 to mid-January 2023 by using convenient and snowball sampling techniques. As this generation is primarily in school, college, or university (Ponsree et al. 2023), the directed link address to Google Forms was sent to these sample groups. With the snowball sampling technique, researchers requested at the end for participants to further forward the link to their friends if they were comfortable.

Additionally, the recommended sample size when applying the SEM method is 250 to 500 because the probability of unexpected issues during data analysis is low (Bentler and Chou 1987). The data were collected from 400 participants. After the data-clearing process, there were 390 valid participants.

*3.2. Research Design and Instrument*

The questionnaire was developed to analyze behavioral data, constructs, and suggestions related to the use of m-banking services. In this study, a five-point Likert scale was adopted into a questionnaire design which was suggested by Joshi et al. (2015) to measure attitude and preferences, as well as the level of influence. The scale was presented from numbers 1 (lowest impact) to 5 (highest impact). Within the questionnaire, there were four main sections inclusive of general information about participants, background information about using m-banking services, the questions on each factor relative to m-banking based on the extension of the TAM and UTAUT theories, and the suggestion. The constructs and scaled items are shown in Table A1. Due to a weakening in the normality assessment of the first item of perceived usefulness (PU1), the item was excluded from the analysis. Additionally, Cronbach's alpha coefficient ($\alpha$) of each construct has been validated. A value

exceeding 0.7 indicates high reliability (Makhoul 1975). Table A2 exhibits that all constructs demonstrated a high-reliability level and represented the usability of the questionnaire.

## 4. Research Findings and Discussion

Within this section, the statistical data are explained and discussed inclusive of the participants' characteristics, normality testing, confirmatory factor analysis (convergent validity, common method bias, and discriminant validity), structural equation modeling, and path analysis.

### 4.1. Characteristics of the Participants

The data were collected from Generation Z aged between 18 and 25 years old who are using m-banking payments. The majority were female (65.13%), followed by male (27.69%). The main participants were students (89.74%). They had a monthly income lower than THB 10,000 (70.77%), consequently, most of them have monthly expenses of less than THB 10,000 (81.79%). About one-third (35.64%) were heavy users, which were those using m-banking payments over twenty times a month, and another one-third (34.88%) used them less than five times a month.

### 4.2. Test of Normality

To assess the normality of the collected data, mean, standard deviation (s.d.), skewness, and kurtosis were analyzed using the SPSS Statistics program. This type of normality testing is widely used in online payment acceptance studies (Ponsree et al. 2023; Mufarih et al. 2020; Sun et al. 2021; Yan et al. 2021; Parikh et al. 2021; Ponsree et al. 2021), is best described as the characteristics of the data distribution, and is understandable with lesser effort (Mardia 1970; Liu et al. 1999). The mean of the data ranged from 3.50 (PR4) to 4.69 (PU2). The s.d. of the constructs ranged from 0.596 (PU construct) to 1.035 (PT construct). The symmetry deviation of the collected data was measured by the skewness and kurtosis tests (Mardia 1970; Liu et al. 1999). The value of skewness between −2 and 2 and the value of skewness between of −7 and 7 illustrated the normality of data distribution (West et al. 1995). The left-skewed distribution was represented by a negative value, while the right-skewed distribution was represented by a positive value. A negative value of kurtosis illustrated flat distribution, while a positive value indicated peaked distribution. Therefore, the data of the current study were considered normally distributed because the skewness value of the constructs ranged from −1.839 (PU construct) to −0.599 (PR construct), and the kurtosis value of the constructs ranged from −0.028 (PT construct) to 3.207 (PU construct).

### 4.3. Confirmatory Factors Analysis

CFA was the method of testing the validity of the research model, the first step before the SEM examination (Anderson and Gerbing 1988). CFA has been used to validate the relationship between latent variables or constructs (unobserved variables) and observed variables, which is a fundamental step of SEM (Ullman and Bentler 2003; Anderson and Gerbing 1988). The fitness of the overall CFA model must meet the threshold, including a chi-square value or $x^2$ ($p < 0.05$), CMIN/df < 3.00, RMR < 0.08, AGFI > 0.80, PGFI > 0.50, IFI > 0.90, TLI > 0.90, CFI > 0.90, and RSMEA < 0.08 (Hair et al. 2013). The model fit indices of the current overall CFA model achieved the suggested values: $x^2$ = 678.842 ($p < 0.001$), CMIN/df = 2.108, RMR = 0.022, AGFI = 0.862, PGFI = 0.706, IFI = 0.956, TLI = 0.947, CFI = 0.955, and RSMEA = 0.053.

#### 4.3.1. Convergent Validity

To test the relationship within and among latent variables and observed variables, the convergent validity was evaluated, which is inclusive of: factor loading ($\lambda$), which shows the internal relationships within a factor or construct. The preferable value of factor loading should higher than 0.50 (Hair et al. 2013); the average variance extract (AVE), which is the estimation of latent variable's variance in the relation to measurement error and for which

the threshold value should be higher than 0.50, representing a good relation (Fornell and Larcker 1981); and composite reliability (CR), which indicates the internal consistency of a particular construct and items used (questions). A strong internal relationship requires a high value of CR, which should be greater than 0.70 (Hair et al. 2013). As exhibited in Table A2, all measurement items achieved the threshold. Thus, the convergent validity is satisfied.

### 4.3.2. Common Method Bias

The current study tested the common method bias (CMB) by using the multitrait–multimethod (MTMM) model (Campbell and Fiske 1959). The MTMM technique enables a more comprehensive investigation of CMB by designing at least two traits with several methods (constructs) for the analysis (Kyriazos 2018). The eight constructs, namely, PU, PEU, SIE, FC, PT, PR, BI, and AU, were fully correlated, and three traits (eight methods x three traits) were nested into the CMB analysis. The model fitness of the MTMM model (eight methods x three traits) achieved the following threshold: $x^2 = 672.721$ ($p < 0.001$), CMIN/df = 2.096, RMR = 0.023, AGFI = 0.862, PGFI = 0.704, IFI = 0.956, TLI = 0.948, CFI = 0.956, and RSMEA = 0.053 (Hair et al. 2013). To evaluate, the differences between the standardized regression weight from the overall CFA model ($\lambda$) and the standardized regression weight from the MTMM model ($\lambda_{MTMM}$) must be less than 0.2, suggesting that the data are fully free of CMB. As illustrated in Table A2, the differences in the standardized regression weights ($\lambda$—$\lambda_{MTMM}$) represent an unserious issue of CMB.

### 4.3.3. Discriminant Validity

To confirm the uniqueness of latent variables, discriminant validity is an important step to validate by testing the correlations of each pair of latent variables (Henseler et al. 2015). A chi-square difference test is one of the methods of testing discriminant validity among constructs (Bollen 1989; Jöreskog and Sörbom 1989; Segars 1997). The methodology compares the unconstrained and constrained models of each pair of constructs. For the unconstrained model, a pair of constructs allowed freely correlated estimation. In contrast, the correlation coefficient value of each pair of constructs was fixed at 0.5 for the constrained model. When comparing the chi-square and the degree of freedom values, the different value of chi-square should be greater than 3.84, with the different value of degrees of freedom at 1 ($p < 0.05$) (Bollen 1989; Jöreskog and Sörbom 1989; Segars 1997). The 28 pairs of constructs were paired against one another. As exhibited in Table A3, all those pairs achieved the recommended value (Bollen 1989; Jöreskog and Sörbom 1989; Segars 1997). Even though the correlation coefficient value between a pair of PU and PEU (0.889) had the highest value (see Table A3), the chi-square difference test illustrated the difference among them at the model level. A pair of PU and PEU contained a chi-square difference value of 65.259, with a different value of the degree of freedom at 1 ($p < 0.001$). Thus, the discriminant validity of all pairs of constructs was achieved.

### 4.4. Structural Equation Modeling: Path Analysis and Discussion

After validating the capabilities of the measurement models, constructs, and items used, SEM is the second step to examine the crucial effect of path relationships or testing the hypotheses (Ullman and Bentler 2003; Buhi et al. 2007; Anderson and Gerbing 1988). The SEM model was fit with the threshold of all aspects (Hair et al. 2013): $x^2 = 831.709$ ($p < 0.001$), CMIN/df = 2.483, RMR = 0.079, AGFI = 0.845, PGFI = 0.719, IFI = 0.938, TLI = 0.930, CFI = 0.938, and RSMEA = 0.062.

According to Figure 1 and Table 1, the results of the SEM path analysis supported fourteen out of fifteen hypotheses. The standardized coefficient or coefficient beta (β) was used to discuss the importance of a particular path relationship by using its absolute value (Vogt 2005). There was a confirmation of H1: perceived usefulness had a significant effect on behavioral intention to use m-banking (β = 0.518, *t*-value = 2.587, *p* < 0.05). This result is consistent with the prior information system study by Davis (1989) and is in line with the recent m-banking study by Karim et al. (2020).

Regarding H2a, the outcome confirmed the effect of perceived ease of use on perceived usefulness, and the result also illustrated the strongest effect among all of the path relationships (β = 0.809, *t*-value = 14.743, *p* < 0.001). Concerning the usefulness of m-banking, easy processes enhance the usefulness of the m-banking system. Moreover, a different contextual investigation concluded that perceived ease of use was a key indicator determining perceived usefulness. For instance, Sitorus et al. (2019) illustrated a robust effect among these two factors in the Indonesian context, Karim et al. (2020) found the greatest effect of perceived ease of use on the usefulness of smartphone banking applications among millennials in the context of Malaysia, and Malaquias and Silva (2020) also found the strongest influence of such an effect, though in a rural area of Brazil. However, H2b, the influence of perceived ease of use on the behavioral intention to use m-banking, cannot be confirmed by the current study (β = −0.189, *t*-value = -0.993, *p* = 0.321). This finding is contrary to the antecedent researches by Davis (1989), Karim et al. (2020), and Van et al. (2021), who found the significant effect of perceived ease of use on behavioral intention. On the other hand, in the UTAUT2-based theory, Owusu Kwateng et al. (2019) and Thusi and Maduku (2020) also found an insignificant effect of effort expectancy on behavioral intention to use m-banking. Because the users of a particular technology are familiar with the functionalities and system operations, they use the technology with lower effort than inexperienced individuals (Liébana-Cabanillas et al. 2018). Furthermore, Generation Z, as digital natives (Oxford Economics 2021), is especially conversant with modern technology. As a result, they would process the technology with less effort in accordance with their primary characteristics.

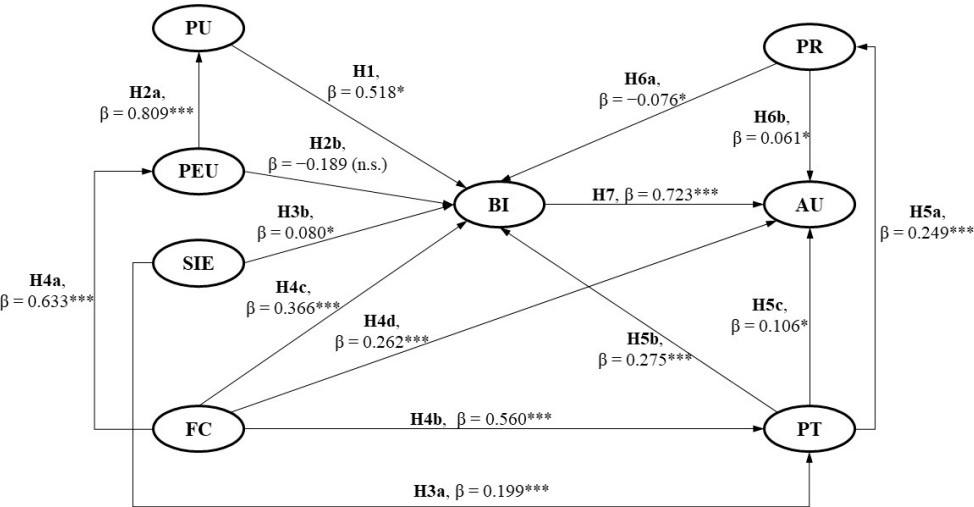

**Figure 1.** The results from SEM path analysis. Denotes that *—*p* < 0.05 and ***—*p* < 0.001.

Based on the findings, there was a confirmation of H3a and H3b: social influence had a direct impact on perceived trust (β = 0.199, *t*-value = 5.328, *p* < 0.001) and behavioral intention (β = 0.080, *t*-value = 2.461, *p* < 0.05), respectively. Another study also supported this relationship among Generation Z (Wei et al. 2021) and among m-banking users (Zhou et al. 2010). Especially in Thailand, Generation Z is more collaborative, relies on, and trusts friends and family (Farrell and Phungsoonthorn 2020). In the same vein, Saprikis et al. 2022) supported that the behavioral intention of both m-banking users and nonusers in

Greece was affected by social influence. The positive relationship of social influence on behavioral intention to use m-banking has also been confirmed in many countries, e.g., Pakistan (Farah et al. 2018), Maldives (Parayil Iqbal et al. 2022), and Sri Lanka (Samsudeen et al. 2022). Notwithstanding this, some researchers cannot confirm an effect (Thusi and Maduku 2020; Merhi et al. 2021; Owusu Kwateng et al. 2019).

The current study demonstrated an intense influence of facilitating conditions towards perceived ease of use. H4a was accepted ($\beta$ = 0.633, *t*-value = 10.765, $p < 0.001$). The availability of necessary resources for learning the way to use m-banking was crucial for m-banking facility providers to improve and maintain updating (Oliveira et al. 2016). Additionally, H4b was supported: the availability of resources and ease of access can generate consumers' perceived trust ($\beta$ = 0.560, *t*-value = 8.499, $p < 0.001$). Moreover, the result confirmed H4c, which hypothesized that facilitating conditions directly impacted behavioral intention ($\beta$ = 0.366, *t*-value = 4.318, $p < 0.001$). Saprikis et al. 2022) discussed that the intention of m-banking application users was considerably influenced by facilitating conditions, but the nonuser's intention was not. In the moderation of age and gender comparisons, Merhi et al. (2021) unveiled that Lebanese m-banking users' intention was positively influenced by facilitation conditions regardless of age and gender. Parayil Iqbal et al. (2022) and Samsudeen et al. (2022) also illustrated a significant effect of facilitating conditions on behavioral intention. Furthermore, H4d was also supported by the research outcome ($\beta$ = 0.262, *t*-value = 4.058, $p < 0.001$). The confirmation of the positive direct effect of facilitating conditions on actual usage is consistent with Venkatesh et al. (2003, 2012), who originally formulated the UTAUT and UTAUT2, respectively. Likewise, a similar study of m-banking visualized a similar effect in Saudi Arabia (Baabdullah et al. 2019) and South Africa (Thusi and Maduku 2020).

**Table 1.** Path analysis results. Denotes that *—$p < 0.05$ and ***—$p < 0.001$.

| Hypotheses | | Path Relationship | Standardized Coefficient ($\beta$) | *t*-Value | *p*-Value | Results |
|---|---|---|---|---|---|---|
| H1 | | PU → BI | 0.518 | 2.587 | 0.010 * | Supported |
| H2 | H2a | PEU → PU | 0.809 | 14.743 | *** | Supported |
| | H2b | PEU→ BI | −0.189 | −0.993 | 0.321 | Not supported |
| H3 | H3a | SIE → PT | 0.199 | 5.328 | *** | Supported |
| | H3b | SIE → BI | 0.080 | 2.461 | 0.014 * | Supported |
| H4 | H4a | FC → PEU | 0.633 | 10.765 | *** | Supported |
| | H4b | FC → PT | 0.560 | 8.499 | *** | Supported |
| | H4c | FC → BI | 0.366 | 4.318 | *** | Supported |
| | H4d | FC → AU | 0.262 | 4.058 | *** | Supported |
| H5 | H5a | PT → PR | 0.249 | 3.391 | *** | Supported |
| | H5b | PT → BI | 0.275 | 5.220 | *** | Supported |
| | H5c | PT → AU | 0.106 | 2.150 | 0.032 * | Supported |
| H6 | H6a | PR → BI | −0.076 | −2.408 | 0.016 * | Supported |
| | H6b | PR → AU | 0.061 | 2.002 | 0.045 * | Supported |
| H7 | | BI → AU | 0.723 | 10.655 | *** | Supported |

Moreover, the extension of perceived trust in the TAM and UTAUT models demonstrated significant relationships. H5a, which hypothesized that perceived trust has a negative effect on perceived risk, was supported ($\beta$ = 0.249, *t*-value = 3.391, $p < 0.001$). The more trust created in the customer demolishes perceived risk (Fortes and Rita 2016; Jarvenpaa et al. 2000). In other words, a high confidence in mobile banking can mitigate the concerns. Additionally, there was a confirmation of H5b, that the perceived trust positively and significantly influenced behavioral intention to use m-banking ($\beta$ = 0.275, *t*-value = 5.220, $p < 0.001$) (Oliveira et al. 2014; Gu et al. 2009; Liébana-Cabanillas et al. 2014b; Liébana-Cabanillas et al. 2018; Liebana-Cabanillas et al. 2020). Karim et al. (2020) exhibited that millennials in Malaysia perceived trust in m-banking applications and Parayil Iqbal et al. (2022) determined perceived trust as a key determinant of behavioral intention to adopt m-banking in the Maldives. In contrast, Farah et al. (2018) were unable to corroborate

perceived trust in the behavioral intention to adopt m-banking among people in Pakistan. In the case of H5c, perceived trust illustrated a positive effect on actual usage ($\beta$ = 0.106, *t*-value = 2.150, $p < 0.05$). In recent times, a group of researchers further examined the effect of perceived trust on use behavior or actual usage in addition to behavioral intention (Malaquias and Silva 2020; Owusu Kwateng et al. 2019). Additionally, Malaquias and Silva (2020) supported a good fit of the influence of perceived trust on actual usage among farmers in rural areas in Brazil.

On the other hand, H6a and H6b were significantly supported: the integration of perceived risk in the TAM and UTAUT theories found a negative effect on behavioral intention ($\beta$ = −0.076, *t*-value = −2.408, $p < 0.05$) and actual usage ($\beta$ = −0.061, *t*-value = 2.002, $p < 0.05$), respectively. In the Vietnamese context, Van et al. (2021) integrated risk-based analysis with the TAM theory and found that perceived risk, a second-order factor comprising financial risk, social risk, time risk, privacy risk, security risk, and performance risk, has a negative effect on m-banking intention among commercial bank consumers. In addition, Thusi and Maduku (2020) also revealed a negative impact of perceived risk on use behavior among millennial banking customers from five South African retail banks (Absa, Capitec, First National Bank, Nedbank, and Standard Bank).

Lastly, in H7, the findings also confirmed a robust relationship between behavioral intention and actual usage ($\beta$ = 0.723, *t*-value = 10.655, $p < 0.001$). This finding is in line with antecedent information system studies (Oliveira et al. 2014; Zhou et al. 2010; Venkatesh et al. 2003, 2012). A plethora of scholars who conducted the m-banking acceptance found a repetitive strong effect of behavioral intention on use behavior (Farah et al. 2018; Owusu Kwateng et al. 2019; Thusi and Maduku 2020; Merhi et al. 2021; Le et al. 2020) and actual usage (Parayil Iqbal et al. 2022).

## 5. Research Implications

### 5.1. Theoretical Implications

The current study uncovered Generation Z's behavior towards the use of m-banking during the unprecedented time of COVID-19 in Thailand. The research model contributed to the TAM theory (Davis 1989) and the UTAUT theory (Venkatesh et al. 2003). The recent research framework has integrated perceived ease of use and perceived usefulness from the TAM theory, social influence and facilitating conditions from the UTAUT theory, and perceived trust and perceived risk to assess the behavioral intention to use m-banking and actual usage. The effect of perceived ease of use on behavioral intention was contrary to the findings of Davis (1989) and Venkatesh et al. (2003). A characteristic of Generation Z is that they prefer digital technology and are familiar with new technology (Oxford Economics 2021). They need less time to learn the way to process technology when compared with the elder generational cohorts (Oxford Economics 2021). An assessment of the direct effect of perceived ease of use on perceived usefulness exhibited that it has the strongest relationship among the tested hypotheses. This sheds light on the hidden relationship in the TAM model. Similarly, the effect of effort expectancy and performance expectancy from the UTAUT theory was significant, of which the influence was proven by a previous study under the investigation of m-banking acceptance (Saprikis et al. 2022). Additionally, we highlighted the impact of social influence on perceived trust, the effect of facilitating conditions on perceived ease of use, and the impact of facilitating conditions on behavioral intention beyond the original UTAUT model. Even though facilitating conditions and behavioral intention were conceptualized within the UTAUT and UTAUT2 models, Venkatesh et al. (2003, 2012) excluded their effect from the information system analysis. However, recent studies of m-banking acceptance further investigated the aforementioned effects beyond user behavior (Boonsiritomachai and Pitchayadejanant 2017; Saprikis et al. 2022; Owusu Kwateng et al. 2019; Farah et al. 2018; Thusi and Maduku 2020; Merhi et al. 2021; Parayil Iqbal et al. 2022; Samsudeen et al. 2022). The extension of perceived trust and perceived risk into the study helped to better understand Generation Z's perception of online environments, particularly of m-banking. The positive effect of perceived trust ($\beta$ = 0.275) was

greater than the negative effect of perceived risk ($\beta = -0.076$) on the behavioral intention to use m-banking among Generation Z in Thailand. In conclusion, the present m-banking acceptance study extended the understanding of Generation Z by various potential factors from the TAM and UTAUT theories during the COVID-19 pandemic, in which perceived trust and perceived risk of m-banking have been taken into account to better understand digital technology acceptance.

### 5.2. Practical Implications

Generation Z is the digital native generation that will play a great role in the future of the e-commerce market. Additionally, the trend of using mobile payment is the median for e-commerce in the Asia Pacific Region and is growing significantly (WorldPay 2020; PwC 2020). This research paper would impact many parties. First, knowing their behavior and intention would ensure that m-banking facility providers can provide up-to-date payment solutions based on their preferences. For example, (i) designing a user-friendly and multi-purpose application, (ii) promoting the m-banking application with trusted and well-known influencers that would enhance their trust-building, and (iii) providing and illustrating the method of using the applications and services available online with ease of access. Second, the m-banking business adopters operating both click-and-mortar services would gain the advantages of such digital services because people are avoiding touching things during COVID-19 to reduce the spread of infection (Smith and Sammer 2021). Therefore, m-banking is a clever alternative payment rather than physical cash. Third, governments, especially of developing countries, should launch the relevant policies pushing a digital environment and building a good digital payment infrastructure, and be ready for a digital transformation era that is significantly driven by the digital native generation. Finally, Generation Z itself would benefit from this present study when all those first three parties responded and offered various alternatives of payment solutions based on individual preferences. The variety of alternatives leads to comparable benefits and transaction costs.

In respect of communication arts and current discoveries, there were some suggestions to consider. Due to Generation Z being acknowledged as digital natives and having a high level of digital competence, advertisement agencies, promoters, content creators, or equivariant might consider these following factors: first, as per the result, perceived usefulness was the strongest predictor of behavioral intention to use m-banking services ($\beta = 0.518$). They should emphasize the helpful functionalities, the purposeful mobile banking applications, multi-purpose services, and time saving in the storyboard and content; in particular, they should especially emphasize easy transactional processes, since they significantly increase Generation Z's perceived usefulness ($\beta = 0.809$). Second, facilitating conditions were the predictor that had a significant impact on perceived ease of use, perceived trust, behavioral intention, and actual usage. If potential problems arise throughout the application process, the promoters should present them with alternative solutions, such as the online chatbot, official website, social media channels, or other necessary fundamental guidebook sources. Third, as mentioned in the preceding paragraph, perceived trust was also important in influencing behavioral intention and actual usage, and it was the component that alleviated consumers' perceived risk and concern. Content creators are expected to collaborate with the mobile application facility providers in developing a statement that potentially increases the perceived trust and the confidence of the consumers, and that further communicates such a message or statement to them. In parallel, trustworthy brand ambassadors and influencers would shape customers' intention and usage. Finally, the quality of contents and visualization can be strengthened when friends, family, and relatives are included since their perceived trust and intention are altered. Creators of contents and storyboards may be concerned about the characteristics of Generation Z and adapt the theme, communication message, composition, and environment to their preferences.

### 6. Conclusions

As the digital transformation of Thailand has been accelerated by the COVID-19 pandemic, digital acceptance studies in different contexts and digital tools are essential (Srisathan and Naruetharadhol 2022). The current study has taken this investigation post COVID-19. For the conceptualization, the relationships between trust and risk perception of Generation Z were examined by extending them with the TAM and UTAUT theories. When comparing the coefficient beta of each factor that affects behavioral intention with the use of m-banking, the results uncovered that perceived usefulness ($\beta = 0.518$) has the strongest effect, following by facilitating conditions ($\beta = 0.366$) and perceived trust ($\beta = 0.275$), respectively. In other words, perceived usefulness was a key indicator to determine Generation Z's intention to use m-banking in Thailand. Nonetheless, the strongest determinant among all path relationships was the effect of perceived ease of use on perceived usefulness ($\beta = 0.809$), which could imply that once Generation Z customers find m-banking is easy to use, they tend to feel that it is useful and increases their job-performance easily. Notwithstanding this, perceived ease of use had no direct effect on behavioral intention ($p = 0.321$). On the other hand, perceived risk has a low negative effect on behavioral intention and actual usage when compared with perceived trust, which has a stronger positive effect. Additionally, perceived trust has a negative impact on perceived risk; the more customers perceived trust, the perception of risk of m-banking usage reduces. Based on the findings, it can be concluded that Generation Z has higher confidence and lower concern about mobile banking usage when comparing the $\beta$ value. The $\beta$ vale of the path PT → BI was 0.275, PT → AU was 0.106, PR → BI was −0.076, and PR → AU was 0.061. This present study would benefit and be advantageous to the government sectors in developing countries, m-banking facility providers, advertisement agencies, content creators, both click-and-mortar business operations, as well as Generation Z and general customers. There are some limitations to this paper. This research is limited to Generation Z m-banking users, lacks comparison data with Generation Z non-users, and is incomparable with data from other generational cohorts. Future research may study broader populations with a comparison of different generations using m-banking services. Moreover, future research may identify other possible factors that might affect customer intention and actual usage.

**Author Contributions:** Conceptualization, K.P. and W.S.; methodology, K.P.; software, P.N.; validation, K.P., P.N. and W.S.; formal analysis, K.P.; investigation, K.P.; resources, W.S.; data curation, W.S.; writing—original draft preparation, K.P.; writing—review and editing. All authors have read and agreed to the published version of the manuscript.

**Funding:** This research was funded by KKUIC Reseach Fund.

**Institutional Review Board Statement:** Khon Kaen University Ethics Committee for Human Research, Khon Kaen University, Khon Kaen, Thailand, has made an agreement that this study has met the criteria of the Exemption Determination Regulations (HE653317).

**Informed Consent Statement:** Informed consent was obtained from all subjects involved in the study.

**Data Availability Statement:** Data is unavailable due to privacy or ethical restrictions.

**Conflicts of Interest:** The authors declare no conflict of interest.

## Appendix A

**Table A1.** Questions used in the questionnaire survey.

| Constructs | Items (Questions) | Adapted From |
|---|---|---|
| PU | PU1: Using m-banking helps me do my transaction more quickly.<br>PU2: Using m-banking helps me do my transaction easily.<br>PU3: Using m-banking would save my time for online transaction activities.<br>PU4: I found that using m-banking is useful for my online transaction. | (Davis 1989) |
| PEU | PEU1: M-banking is easy for me to learn how to use it.<br>PEU2: I can quickly use the m-banking method.<br>PEU3: I found the m-banking system easy to proceed with.<br>PEU4: I found m-banking easy to use overall. | (Davis 1989) |
| SIE | SIE1: People who are close to me think that I should use m-banking.<br>SIE2: People who influence my behavior think that I should use m-banking.<br>SIE3: People whose opinions I value prefer that I use m-banking. | (Venkatesh et al. 2003, 2012) |
| FC | FC1: I have the resources necessary to use m-banking.<br>FC2: I know what is necessary to use the m-banking method.<br>FC3: I can get help from others when I have difficulties using m-banking. | (Venkatesh et al. 2003, 2012) |
| PT | PT1: M-banking is trustworthy.<br>PT2: M-banking is reliable.<br>PT3: The process of m-banking is secure.<br>PT4: The m-banking provider will maintain terms and commitments strictly. | (Pavlou 2003; Oliveira et al. 2014; Kim et al. 2009) |
| PR | PR1: Using m-banking for online transactions is a risky choice.<br>PR2: Providing my personal information on m-banking activity is a risky choice.<br>PR3: Others may know the information about my online transaction via m-banking.<br>PR4: M-banking is insecure. | (Schlosser et al. 2006) |
| BI | BI1: Assuming I have used m-banking, I would intend to use it as my payment method.<br>BI2: I intend to use an m-banking rather than any payment method.<br>BI3: If I use an m-banking method, I believe that I would use it as my primary payment.<br>BI4: I intend to use the m-banking service regularly in the future. | (Oliveira et al. 2014; Kim et al. 2009) |
| AU | AU1: I often use an m-banking platform.<br>AU2: I often use m-banking to transfer money.<br>AU3: I often use m-banking to make payments. | (Oliveira et al. 2014; Venkatesh et al. 2003, 2012) |

**Table A2.** Construct validity.

| Constructs | Scale Items | $\alpha$ | $\lambda$ | AVE | CR | $\lambda_{MTMM}$ | $\lambda - \lambda_{MTMM}$ |
|---|---|---|---|---|---|---|---|
| PU | PU2<br>PU3<br>PU4 | 0.824 | 0.788<br>0.768<br>0.787 | 0.610 | 0.824 | 0.788<br>0.769<br>0.787 | \|0.000\|<br>\|0.001\|<br>\|0.000\| |
| PEU | PEU1<br>PEU2<br>PEU3<br>PEU4 | 0.898 | 0.830<br>0.828<br>0.805<br>0.853 | 0.688 | 0.898 | 0.831<br>0.827<br>0.805<br>0.850 | \|0.001\|<br>\|0.001\|<br>\|0.000\|<br>\|0.003\| |
| SIE | SIE1<br>SIE2<br>SIE3 | 0.911 | 0.864<br>0.868<br>0.906 | 0.774 | 0.911 | 0.863<br>0.868<br>0.907 | \|0.001\|<br>\|0.000\|<br>\|0.001\| |
| FC | FC1<br>FC2<br>FC3 | 0.832 | 0.838<br>0.833<br>0.701 | 0.629 | 0.835 | 0.838<br>0.834<br>0.703 | \|0.000\|<br>\|0.001\|<br>\|0.002\| |

**Table A2.** *Cont.*

| Constructs | Scale Items | $\alpha$ | $\lambda$ | AVE | CR | $\lambda_{MTMM}$ | $\lambda—\lambda_{MTMM}$ |
|---|---|---|---|---|---|---|---|
| PT | PT1 | 0.899 | 0.896 | 0.702 | 0.903 | 0.896 | \|0.000\| |
| | PT2 | | 0.906 | | | 0.907 | \|0.001\| |
| | PT3 | | 0.820 | | | 0.821 | \|0.001\| |
| | PT4 | | 0.716 | | | 0.711 | \|0.005\| |
| PR | PR1 | 0.890 | 0.855 | 0.673 | 0.892 | 0.855 | \|0.000\| |
| | PR2 | | 0.836 | | | 0.836 | \|0.000\| |
| | PR3 | | 0.817 | | | 0.817 | \|0.000\| |
| | PR4 | | 0.772 | | | 0.772 | \|0.000\| |
| BI | BI1 | 0.928 | 0.811 | 0.766 | 0.929 | 0.811 | \|0.000\| |
| | BI2 | | 0.895 | | | 0.894 | \|0.001\| |
| | BI3 | | 0.891 | | | 0.890 | \|0.001\| |
| | BI4 | | 0.900 | | | 0.897 | \|0.003\| |
| AU | AU1 | 0.829 | 0.783 | 0.631 | 0.837 | 0.783 | \|0.000\| |
| | AU2 | | 0.776 | | | 0.777 | \|0.001\| |
| | AU3 | | 0.824 | | | 0.825 | \|0.001\| |

**Table A3.** Bivariate correlation and chi-square difference test. Notes that: ***—$p < 0.001$, constructs are different at the model level.

| Constructs | Bivariate Correlation | Unconstrained | | Constrained | | $X^2$ Differences (with d.f. Different at 1) |
|---|---|---|---|---|---|---|
| | | $X^2$ | d.f. | $X^2$ | d.f. | |
| PU and PEU | 0.889 | 29.397 | 13 | 94.656 | 14 | 65.259 *** |
| PU and SIE | 0.381 | 8.455 | 8 | 98.254 | 9 | 89.799 *** |
| PU and FC | 0.650 | 8.949 | 8 | 104.953 | 9 | 96.004 *** |
| PU and PT | 0.441 | 29.950 | 13 | 148.868 | 14 | 118.918 *** |
| PU and PR | 0.294 | 41.951 | 13 | 131.398 | 14 | 89.447 *** |
| PU and BI | 0.571 | 70.759 | 13 | 137.752 | 14 | 66.993 *** |
| PU and AU | 0.699 | 21.495 | 8 | 91.410 | 9 | 69.915 *** |
| PEU & SIE | 0.391 | 39.366 | 13 | 105.526 | 14 | 66.160 *** |
| PEU and FC | 0.654 | 28.007 | 13 | 97.538 | 14 | 69.531 *** |
| PEU and PT | 0.466 | 37.073 | 19 | 125.592 | 20 | 88.519 *** |
| PEU and PR | 0.213 | 45.135 | 19 | 133.389 | 20 | 88.254 *** |
| PEU and BI | 0.543 | 60.694 | 19 | 110.549 | 20 | 49.855 *** |
| PEU & AU | 0.611 | 34.944 | 13 | 95.130 | 14 | 60.186 *** |
| SIE and FC | 0.527 | 5.426 | 8 | 53.225 | 9 | 47.799 *** |
| SIE and PT | 0.459 | 29.683 | 13 | 192.059 | 14 | 162.376 *** |
| SIE and PR | 0.183 | 27.087 | 13 | 82.636 | 14 | 55.549 *** |
| SIE and BI | 0.446 | 32.150 | 13 | 60.290 | 14 | 28.140 *** |
| SIE and AU | 0.436 | 22.660 | 8 | 66.494 | 9 | 43.834 *** |
| FC and PT | 0.563 | 26.586 | 13 | 78.434 | 14 | 51.848 *** |
| FC and PR | 0.291 | 37.280 | 13 | 92.250 | 14 | 54.970 *** |
| FC and BI | 0.640 | 34.572 | 13 | 57.618 | 14 | 23.046 *** |
| FC and AU | 0.696 | 48.331 | 8 | 83.897 | 9 | 35.566 *** |
| PT and PR | 0.186 | 37.064 | 19 | 111.910 | 20 | 74.846 *** |
| PT and BI | 0.586 | 48.725 | 19 | 76.739 | 20 | 28.014 *** |
| PT and AU | 0.622 | 30.572 | 13 | 70.862 | 14 | 40.290 *** |
| PR and BI | 0.116 | 37.533 | 19 | 112.039 | 20 | 74.506 *** |
| PR and AU | 0.222 | 52.402 | 13 | 123.654 | 14 | 71.252 *** |
| BI and AU | 0.855 | 68.806 | 13 | 90.398 | 14 | 21.592 *** |

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
