# Peer review of "The Confidence of and Concern about Using Mobile Banking among Generation Z: A Case of the Post COVID-19 Situation in Thailand"

_socsci, doi:10.3390/socsci12040198_

Round 1

Reviewer 1 Report

The article is interesting, raises important and current issues. It is prepared carefully and shows a considerable amount of research work. Despite its undoubted advantages, authors should pay attention to several aspects. Here they are:

The introduction does not describe the purpose of the work, methods and organisation of the research process.

In the introduction, the authors should expand the characteristics of generation Z. The issue is treated very briefly in the introduction, although this group is covered throughout the article.

Authors should justify why they study only Generation Z.

Since the authors emphasise the confidence and concern in the title of the article, they should devote more space to these terms in the introduction and in the article itself.

The authors should refer more broadly to the representativeness of the sample - most of the respondents are students, i.e. they are assumed to have a higher level of knowledge in the field of digitization and banking than non-students at a young age. How were the respondents selected? When was the survey conducted, how long did it take to collect the questionnaires.

Authors should address why they did not use other types of normality testing methods or tests (e.g. Shapiro-Wilk).

Did the authors test the coherence of the survey, e.g. using Crombach's alpha coefficient, or did they conduct a pilot study?

The article contains some grammatical errors. Recommended reading by a native speaker.

Author Response

Dear reviewers,

Thank you very much for reviewing the paper “The Confidence of and Concern about Using Mobile Banking Among Generation Z: A case of the Post COVID-19 in Thailand”. We greatly appreciate the reviewers for their constructive comments and suggestions. We have completed the adjustment that the reviewers recommended and addressed the manuscript accordingly.

We have included a point-by-point response to the reviewers in accordance with the changes described in the revised manuscript. Changes to the text and the table in the manuscript are marked in the comments. We hope that you find our revised satisfactory and that should be of interest to the readership of Social Sciences.

Thank you for your consideration of our revised manuscript.

Best regards,

Comments and Suggestions for Authors

The article is interesting, raises important and current issues. It is prepared carefully and shows a considerable amount of research work. Despite its undoubted advantages, authors should pay attention to several aspects. Here they are:

Response: We appreciate your comments.

The introduction does not describe the purpose of the work, methods and organisation of the research process.

Response: We have further described the research objectives, methods, and the organization of the research process in the introduction in accordance with your review (in lines 105 to 126 on page 3).

In the introduction, the authors should expand the characteristics of generation Z. The issue is treated very briefly in the introduction, although this group is covered throughout the article.

Response: According to your comments, we further explained the characteristics of generation Z in the introduction (in lines 86 to 93 on page 2).

Authors should justify why they study only Generation Z.

Response: As per your suggestion, we have justified the reason for choosing generation Z in the methodology section in lines 327 to 334 on page 7.

Since the authors emphasise the confidence and concern in the title of the article, they should devote more space to these terms in the introduction and in the article itself.

Response: According to your suggestions, we have emphasized confidence and concern in the introduction (in lines 70 to 85 on page 2). Furthermore, we also devoted more space throughout the manuscript, e.g., under the result, discussion, and conclusion sections.

The authors should refer more broadly to the representativeness of the sample - most of the respondents are students, i.e. they are assumed to have a higher level of knowledge in the field of digitization and banking than non-students at a young age. How were the respondents selected? When was the survey conducted, how long did it take to collect the questionnaires.

Response: We agree with your valuable suggestions. Therefore, we are more explained on the representativeness of the sample (in lines 327 to 334 on page 7), the sampling method, the period, and the duration of data distribution (in lines 340 to 345 on page 7).

Authors should address why they did not use other types of normality testing methods or tests (e.g. Shapiro-Wilk).

Response: As per your suggestions, we have addressed the reason for choosing this type of normality testing in line 382 to 387 on page 8.

Did the authors test the coherence of the survey, e.g. using Crombach's alpha coefficient, or did they conduct a pilot study?

Response: We further tested Cronbach’s alpha coefficient in Table A2 on the appendix page and explained in section 3.2 in lines 363 to 366 on page 8.

The article contains some grammatical errors. Recommended reading by a native speaker.

Response: As per your suggestions, our paper has been double-checked.

Reviewer 2 Report

Overall, this is a clear, concise, and well-written manuscript. The introduction is relevant and theory based. Sufficient information about the previous study findings is presented for readers to follow the present study rationale and procedures. The methods are generally appropriate, although English need to check and better to explain all statistical data step by step the research follows.

Author Response

Dear reviewers,

Thank you very much for reviewing the paper “The Confidence of and Concern about Using Mobile Banking Among Generation Z: A case of the Post COVID-19 in Thailand”. We greatly appreciate the reviewers for their constructive comments and suggestions. We have completed the adjustment that the reviewers recommended and addressed the manuscript accordingly.

We have included a point-by-point response to the reviewers in accordance with the changes described in the revised manuscript. Changes to the text and the table in the manuscript are marked in the comments. We hope that you find our revised satisfactory and that should be of interest to the readership of Social Sciences.

Thank you for your consideration of our revised manuscript.

Best regards,

Comments and Suggestions for Authors

Overall, this is a clear, concise, and well-written manuscript. The introduction is relevant and theory based. Sufficient information about the previous study findings is presented for readers to follow the present study rationale and procedures. The methods are generally appropriate, although English need to check and better to explain all statistical data step by step the research follows.

Response: We agree with your recommendations. We double-checked the spelling and English language throughout the manuscript. Moreover, the explanation of statistical data step-by-step was further explained in lines 368 to 371 on page 8.

Reviewer 3 Report

-A good language edit is required. Many language errors were noted, a significant flaw in the paper.

-Golden threat is missing in some instances. Meaning that ideas and arguments overlap or are repeated, which makes understanding the complete "story" difficult. Try and improve linkages. Keep one idea/argument in one paragraph, and link it with the paragraph to follow. The language edit could solve this problem.

-Updated sources are used, which is always good.

-Be consistent in the terms used. For example, is it mobile banking, mobile payment, or digital payment service? There is a difference. Although mobile payment is a function of mobile banking, the purpose of this study relates to mobile banking, not mobile payment. Make sure you distinguish.

-Your study would make a more valuable contribution if new constructs in the mobile banking context were tested, as the theories tested in this paper are well-known and tested in the mobile banking context many times before. Merely extending the model does not necessarily make it new. You do, however, make a contribution given the sample that was used.

The constructs and their inter-relationships are well explained.

Generally in SEM, we refer to constructs as factors.

Author Response

Dear reviewers,

Thank you very much for reviewing the paper “The Confidence of and Concern about Using Mobile Banking Among Generation Z: A case of the Post COVID-19 in Thailand”. We greatly appreciate the reviewers for their constructive comments and suggestions. We have completed the adjustment that the reviewers recommended and addressed the manuscript accordingly.

We have included a point-by-point response to the reviewers in accordance with the changes described in the revised manuscript. Changes to the text and the table in the manuscript are marked in the comments. We hope that you find our revised satisfactory and that should be of interest to the readership of Social Sciences.

Thank you for your consideration of our revised manuscript.

Best regards,

Comments and Suggestions for Authors

-A good language edit is required. Many language errors were noted, a significant flaw in the paper.

Response: A

-Golden threat is missing in some instances. Meaning that ideas and arguments overlap or are repeated, which makes understanding the complete "story" difficult. Try and improve linkages. Keep one idea/argument in one paragraph, and link it with the paragraph to follow. The language edit could solve this problem.

Response: According to your suggestions, we have improved the language and edited the errors.

-Updated sources are used, which is always good.

Response: We sincerely thank you.

-Be consistent in the terms used. For example, is it mobile banking, mobile payment, or digital payment service? There is a difference. Although mobile payment is a function of mobile banking, the purpose of this study relates to mobile banking, not mobile payment. Make sure you distinguish.

Response: We agree with your comments. Then, we double-check the terminology throughout the paper. However, we maintain the term from the previous research articles which differently study on mobile banking, mobile payment, or digital payment.

-Your study would make a more valuable contribution if new constructs in the mobile banking context were tested, as the theories tested in this paper are well-known and tested in the mobile banking context many times before. Merely extending the model does not necessarily make it new. You do, however, make a contribution given the sample that was used.

Response: We agree with your comments. Your suggestions are valuable for our future research and improvement.

The constructs and their inter-relationships are well explained.

Response: We truly appreciate it.

Generally in SEM, we refer to constructs as factors.

Response: We agree with your comment.
